# The Necropolice Economy: Mapping Biopolitical Priorities and Human Expendability in the Time of COVID-19

**Mark Howard** 

Department of Politics, University of California, Santa Cruz, CA 95064, USA; mrhoward@ucsc.edu

**Abstract:** Necropolitics centers on the dark side of biopolitics, but if we are to take seriously Jacques Ranciere's reassignment of 'politics' and 'police,' then what is revealed by necropolitical analysis is not simply the capacity to 'make and let die', but also the policing of a contingent order sustained by necropolitics. I describe this process as the necropolice-economy, and in this paper demonstrate its contours with reference to the COVID-19 pandemic which, I argue, has revealed the expendability of particular populations under conditions of risk and uncertainty. My analysis proceeds in three parts. First, I present the thesis of necropolice economy, arguing that the capitalist system has historically produced not simply a political economy, but a policed economy that induces a necropolitics of dispensability for unproductive or replaceable populations. Second, I develop this thesis by examining the relegation of society in relation to the economy amidst the COVID-19 pandemic. Third, I argue that the inability of states to be decisive in the pandemic reveals that the sovereign prerogative to decide on the exception is constrained by capitalist forces. This suggests that the world market is itself a sovereign force, though it is one that remains ever dependent on state violence. To conclude, I ask whether we can channel the trauma of death made visible into processes of memorialization that might catalyze revolutionary action, rather than accelerating the evolution of our necropolice economy into its next capitalist guise—I ask, provocatively, whether an emancipatory necropolitics might yet result from the contemporary moment.

**Keywords:** necropolitics; COVID-19; capitalism; sovereignty

## 1. Introduction

*"It is going to spread further and I must level with you, I must level with the British public: many more families are going to lose loved ones before their time."*

Boris Johnson, UK Prime Minister, March 2020 [1].

*"I'm sorry, some people will die, they will die, that's life . . . [y]ou can't stop a car factory because of traffic deaths."*

Jair Bolsonaro, Brazilian President, March 2020 [2].

Proximal death is revelatory in a way that distant death—death separated from us by time or space—can rarely be. Despite its essential relation to life and consequent omnipresence, for most people, death is only visible when proximate to one's own lifeworld. Martin Heidegger's claim that we have (or had) forgotten the question of the meaning of being, led ultimately to the conclusion we are all beings-towards-death; the phrase memento mori on this account becoming the meaning of being, the decisive factor bestowing our lives with meaning through an urgency of care [3]. The suggestion that we have forgotten this aspect of ourselves is related to the visibility of death and its proximal relation to our own contextual frame: just as a near-death experience brings visibility to our own otherwise invisible (i.e., forgotten) mortality, so too does a death in the family or community make death visible once more—it does so by bringing it into the open, along with factors that led to its representing.

The COVID-19 pandemic, which at the time of writing has contributed to 4.55 million deaths worldwide and has spared almost no community on Earth, has made death visible

to us all. However, in doing so it has also revealed the structural forces behind such an immense scale of death and, for many, near-death experiences. COVID-19 is a traumatic event that has made the world and its obsession with capital strange once again. As a violent intrusion upon social being, it has made death visible at the scale of society and shone a light on the political mechanisms of sovereign power over life and death. However, Achille Mbembe's pathbreaking work *Necropolitics*—summarized neatly as the "power and capacity to dictate who is able to live and who must die"—foregrounds an aspect missed by mainstream analyses of the pandemic crisis: in promoting life, the sovereign power determines not merely who can die, but also who must [4]. COVID-19, in making death visible, in bringing necropolitics into a field of visibility, has also made visible who in our own societies has been rendered expendable and who must necessarily be exposed to death: the elderly, the homeless, racial minorities, immigrants, rural populations [5]; those who are unproductive or whose productivity is so essential that their lives can be given up to the priority of economic continuity.[1]

More specifically then than necropolitics, the COVID-19 pandemic has revealed what commentators inspired by Mbembe's work have been referring to as necroliberalism, necrocapitalism, and necroeconomics [7,8], but which I here refer to as necropolitical economy, for to speak of a necropolitical economy speaks to the utter entanglement between political processes and decisions on the one hand, and economic activities and outcomes on the other. To speak of necroliberalism, necrocapitalism, or necroeconomics is to sideline necropolitics, and this arbitrarily impoverishes debate and analysis. I insist, however, in the argument that follows, to take this move towards necropolitical economy one step further, and to theorize the order-preserving function of the police.[2] Such a move requires not merely a reassignment of the terms police and politics, such that police becomes the determination of normal, acceptable behavior in what we call the political sphere, and politics becomes the term for paroxysms challenging that sphere, for claims from beyond the field of visibility and acceptability for a new order; such a claim also implies that necropolitical economy—as a claim upon the present order made in the name of both the dead who are no longer alive, and the living dead that the police order has deemed expendable for its own self-preservation—may serve as an emancipatory field of action for those otherwise unequally exposed to death. Thus, the crux of my analysis centers on the claim that preceding the possibility of a necropolitical economy—a possibility that has only come into view amidst the COVID-19 crisis—there was, and is, a necropolice economy that functions to preserve the status quo.

The event of COVID-19 is a moment of revelation; by revealing what was already there in the past, it has fostered a vicious debate over the present and future. Will the pandemic simply demonstrate and highlight existing patterns of capitalist accumulation, or might it accelerate them? Might this revelation signal the end of neoliberal capitalism? If so, are we to expect a more just social order[3] or will the future be worse [11]? To answer these questions through an analysis of the necropolice and necropolitical economy, this paper is divided into three parts. First, I present the thesis of necropolice economy, arguing that the capitalist system has historically produced not simply a political economy, but a policed economy that induces a necropolitics of dispensability for unproductive or replaceable populations. Second, I develop this thesis by examining the relegation of society in relation to the economy amidst the COVID-19 pandemic. Finally, I argue that the inability of states to be decisive in the pandemic reveals that the sovereign prerogative to decide on the exception is constrained by capitalist forces. This suggests that the world market is itself sovereign, and that necropolitics must be conjoined with necroeconomics to explain the contemporary moment.

## 2. A Necropolice Economy

The capitalist system as a historically situated mode of production generalized to society as a whole has produced not merely a political-economic function—defined by Foucault as the stage in history whereby the sovereign function begins to determine itself

in terms of domestic efficiency and international competitiveness as opposed to military conquest and war [12]—but also a police-economic function that sustains and preserves the prevailing order of socioeconomic and political relations. To understand this, we must revisit Foucault's critique of Clausewitz, which results in the dictum "politics is a continuation of war by other means".[4] Foucault's argument is that all societies have their origin in conquest and are held together by the continuing exercise of political (generally state) power.[5] Thus, the appearance of peace in society in fact conceals a coded war and otherwise tenuous relationship between diverse groups of individuals under a ruling state apparatus [13,14]. Society, on this account, is the result of some forgotten or ignored historical victory, whereby the boundary separating inside and outside was redrawn, and the population arranged beneath a sovereign authority was asserted anew. Thus, we may advance a preliminary definition of politics as the process of contestation over social form.

However, according to Jacques Rancière, what we typically refer to as politics is not the normal operations of debate over society, the type of occurrences seen in parliaments and congresses, or even in elections and non-revolutionary protests. No, these happenings do not count as politics for they do not in reality contest social form. Rather, these functions occur within the bounds of sanctioned debate; they are permissible, and therefore visible, operations that not only perform the status quo, but serve to legitimate and therefore strengthen it. These functions are part of a hegemonic formation that, for Rancière, are better described as occurring in the domain of the police than in the domain of politics [9].

The various segments of society that are bound by Foucault's politics are therefore actually kept in check by a police function, whereas the act of conquest that brought about this monolithic power structure—typically the state—is the true act of politics. Politics is, as such, a democratic insurgency, a challenge to the status quo; albeit one that—if successful—only produces its own (new) police order.[6] In any given order, the police function serves to actively sustain the unification of society, because remnants of the conquest always persist in the form of social diversity. The reality that society is not unified is what necessitates the police function; however, not only does the police function sustain unity amongst the people so as to form a single society, it also sustains a class hierarchy within that unity to further the ends of the organic whole.

Admittedly, Rancière is not typically associated with the field of biopolitics and has, in the past, explicitly distanced himself from the concept. He argues that a description of society thickened by the dual presence of both politics and police, by acknowledging that portion of society which is excluded from the field of visibility, exceeds the theory of life and its modes of regulation that constitutes biopolitics. For Rancière, it is Foucault's conception of politics that is insufficient in that everything it accounts for is located in the partial and limited field of visibility that Rancière designates as 'police' [15]. Nonetheless, when biopolitics is held up against the notion of necropolitics, its dark and less visible side, Rancière's account regains its appropriateness and acquires a chilling vision of social order.

Thus the first contours of necropolitics become evident, for it is precisely through class hierarchy and the field of visibility that the police order enacts its ability and right to expose certain segments of society to extreme violence and death, perhaps even to reducing them to bare life [7]. In terms of the organic whole, particular groups are, and must be, reduced to a more precarious existence so that the whole of society may persist and even thrive in its current form [16]. In what amounts to a hierarchy of life, this leads in some cases—such as the COVID-19 pandemic—to a sacrificial order [17]. Members of society deemed less visible and less valuable to the organic whole, such as the elderly, unskilled labor (including undocumented workers, particularly in industries such as agriculture), prisoners, the homeless, and indigenous peoples, become expendable [18]. That all of these groups, aside from the first, are disproportionately represented by people of color is not coincidental, but rather speaks to the histories of conquest—the politics—that contemporary society is founded upon.

It is a twisted irony that the disposable are those people most vulnerable to death under exposure. COVID-19 has revealed this to us today, but the decades long stigmatization,

marginalization, abuse, and exposure of HIV/AIDS carriers, mental health patients, and poor folk participating in health studies (e.g., the Tuskegee Syphilis Study) demonstrates this as only part of a historical continuum [17,19]. These, among other members of society, have, in the present pandemic as well as across history, been "subjected to living conditions that confer upon them the status of the living dead" [4]. These are the members of society that Boris Johnson and Jair Bolsonaro refer to in the epigraph of my introduction.

Undoubtedly, Mbembe's original analysis of necropolitics speaks of a more visceral and stark order of violence and exposure to death: apartheid and concentration camps, but the structural violence imposed by the police-economic function also submits to widespread suffering and death, and it is not to be ignored or belittled. The political-economic function of the police order as societal efficiency stands opposed to war and conquest as the domains of violence and death, and yet capitalism is emphatically a necrocapitalism in its ability to bring about violence and death. It is not insignificant that Marx himself was fond of both figurative and literal references to blood and death [20]. His claim that the capitalist must exploit or be destroyed as a capitalist due to the machinations of competition[7] seems trite in the context of Mbembe's necropolitics, and yet it is not only the capitalist whose survival is at stake in this analysis. As Marx carefully documents in *Capital Vol I*, the exploitation that capitalist survival and competition depend on leads to brutal death and suffering for less valued members of society.[8] The necropolitics of Mbembe (discussing, for instance, slavery) just as with the critical analysis of Marx reveals the logic of the capitalist system: everybody has a price. It is all about value, supply, and demand [22].

What distinguishes capitalism from its alternatives is the requirement that it separate and prioritize its own operations from that of the social whole. Whereas socialist modes of production, for instance, take the whole of society as their starting premise, and orient economic policy towards the promotion of social ends, capitalism takes capital as its starting premise and orients economic policy towards capitalist ends. Capitalism undoubtedly absorbs the entirety of society into its functional orbit, yet it employs only that which is necessary for the uninterrupted accumulation of capital; its only necessary means are the means of production. It is this exclusion of any need to promote the health of the social whole that places capitalism at the center of the present analysis, for it is a system that, in the face of mass death, need be concerned only with the continued existence of those segments of society deemed necessary for production, namely: capitalists;[9] skilled and difficult-to-replace workers; and a mass of unskilled easy-to-replace workers backed up by what Marx referred to as a "reserve army of labour" [6]. In the latter case, concrete individuals are deemed expendable provided the abstract whole is not existentially threatened—again, the priority is capitalist accumulation as dictated by market compulsion, and not the reproduction of society as a whole—while the expendability of non-productive members of society has no conditionality[10] and may even be promoted in the aid of preserving society's productive function.[11]

Amidst the COVID-19 pandemic, then, socioeconomic inequality under the class relations of capital have directly contributed to biological—and therefore biopolitical—inequality. The equality we all experience in death is negated by the socioeconomic and political inequality that we experience in life through the necropolitical function of vulnerability. COVID-19, plainly put, does not affect everyone equally. Even when a lockdown is put in place with the ostensible purpose of limiting death, its economic impacts are felt most acutely by those members of society most vulnerable to socioeconomic pressures [19], and these socioeconomic conditions are as much a factor in the mortality of COVID-19 cases—as well as related deaths arising from mental or other health issues—as are the biological factors [18]. More specifically, in the case of COVID-19 infections, comorbidities—the simultaneous presence of two or more diseases in a living being—such as hypertension, diabetes, or coronary heart disease make people more vulnerable to the virus. While admittedly not always the case, these ailments are disproportionately experienced by people living under poor socioeconomic and political conditions; as a result

of where they live, what they can afford to eat, how much exercise their environment and situation allows for, and the levels of stress they experience [23].

### 3. The Necropolitical Economy of COVID-19

> *"Neoliberal capitalism has left in its wake a multitude of destroyed subjects, many of whom are deeply convinced that their immediate future will be one of continuous exposure to violence and existential threat."* [4]

In its role as the driver of internal efficacy and external competitiveness, the state necessarily relegates not merely some sections of society but society as a whole to a status below that of the economy. Foucault claims that the emergence in Germany and the United States of a widespread state-phobia arising in response to the authoritarian dictatorships of Hitler, Mussolini, and Stalin led to what he characterizes as a crisis of governmentality [12]. In response to the crisis, the state's role was formally adjusted such that the sovereign scope of authority became limited to producing and protecting a discrete sphere of economic freedom within which it had—and has—little to no direct control. The sovereign state thus came to be defined—in part at least—by what it is not.[12] Perversely, this leads to the commonsense view that should the state intervene and abuse power in the space of economic freedom, it would have forfeited its own legitimacy under the social contract having failed to protect the rights and freedoms of the population [12]. It will, in this instance, have given up its right to governmental authority. The state effectively acts as a protector of the economy, which is today effectively defined across the globe by capitalist modes of production.

In the service of this economic function, the state sustains and promotes class relations as direct operation of the police function, and under conditions of heightened precarity and the risk of death, such as those conditions produced by COVID-19, the necropolice function this invokes is better categorized as a necropolice economy. The efficiency and competitiveness of the economic sphere as it supports and is supported by the state is dependent on the exposure of less valued members of the population to death and violence.

Much of this has been exacerbated by what commentators such as Tony Sandset refer to as "slow violence", pre-existing social conditions that give rise to endemic rather than epidemic or pandemic risk of death [18]. It is worth remembering on this point that while the necropolice economy has its origin in a necropolitical economy—that is, on a political reconfiguration of social form grounded in the relation of biopolitics and death—necropolice economy is the 'normal' (i.e., enduring) state of social operation. Only now, in a crisis, is the necropolitical-economic function reappearing, for the legitimacy of police function is itself vulnerable to challenge. After all, if this order were based on values acceptable to everyone, policing would not be required and there would be no more need for contestation over social form.[13] However, the finality of this order is illusory: just as necropolitics is a chronic feature of society [18], so too is the necropolice economy. Although the event of COVID-19 would seem to bring onto the scene a necropolice economy to preserve the necropolitical economy, the reverse is true: we were already living in a necropoliced economy and it is now that a political struggle has begun over the contours of the future.

That the prevailing order feels threatened should not be a surprise, for it is caught in a trap. For political leaders such as Boris Johnson, Jair Bolsonaro, and (previously) Donald Trump, COVID-19 is a political crisis of economics and not health. However, even they would be aware that this is a false choice: focusing solely on the economics risks extending or even deepening the health crisis which will in turn lead to greater economic impact. Evidence of looming recession as a result of extreme lockdown policies in Australia and New Zealand are cases in point [24]. Leaders are, as such, damned regardless of their choice of prioritization [25]. In such a situation, it seems perhaps logical, if immoral, to prioritize the economy and economic actors or classes that keep them in office. It is this logic that prompts leaders to treat the elderly as collateral damage while they pursue an ill-fated policy of achieving 'herd immunity' [26], or to take away adequate personal

protective equipment (PPE) from public health workers as a means of reducing public expenditures (which may later be used to justify tax cuts) [27]. However, regardless of such resignation, these political leaders remain trapped in a quandary. Design theorist Benedict Singleton argues that if a trap is "is to be escaped by anything other than luck ... the escapee itself must change: the thing that escapes the trap is not the thing that was caught in it." [28]. The key, for Singleton, is to recognize how one is implicated in the mechanism of entrapment. Herein lies the threat: either the expendable themselves will recognize how they are implicated in mechanism of entrapment—as is well known, the system depends on those it exploits—or the system will, and must, transform itself. Some argue that the latter is already underway and that neoliberal capitalism is transitioning to a libertarian-authoritarian order, one that embraces the prioritization of economic laissez-faire thinking (though not, perhaps, practice) [29].

To go deeper into the rabbit hole of economic prioritization, of necropolitical-economic action, means to negate the popular threat coming from those who have been deemed expendable; to render them invisible once more so as to re-instigate the veneer of social equality [9]. While the homeless and undocumented may not been seen as a significant threat to any prevailing regime, people of color, the elderly, rural populations, and unskilled labor all are. Thus, the necropolitical move in a disaster scenario such as COVID-19 is to make those populations believe they want the economy to be prioritized, to make them believe they are part of the field of equality and that they too can prosper under a police regime of capitalist reproduction [9]. Undoubtedly, the pandemic offers an opportunity for some regions and sectors, which will in turn produce opportunities for members of the population. However, the ideological spin on this will be and has been to greatly inflate the scale of opportunity opening up to subordinated groups in society; this ensures that individual actors—even the most vulnerable—will be led to assess their economic interests in such a way that the potential for a health catastrophe is preferable to cost of public measures that might remove such opportunities [29].

In short, the demos must be deceived. The fact that it was the scale of the global capitalist economy—mass urbanization, frequent international travel, etc.—that exacerbated the crisis in the first place is downplayed; COVID-19 itself is characterized as much as is possible as an agent in itself, its agential quality serving as an alibi for bad or parochial policy decisions [30]; the disposable populations fighting for economic and biological survival are led to believe that their expendable lives perform expendable functions, and that they must accept their conditions, even as the deceit in this claim is revealed by their designation as 'essential' workers; and, by no means finally, mental health issues and young people's life chances are cited as reasons for 'balancing' the disease with economic priorities, even though the reality is that any prioritization of health need not be limited COVID-19 patients, but could and should be expanded to the health needs of society as a whole—needs that could be met if only there was the political will [31]. All this, and more, despite research demonstrating that to prioritize health might actually be a better long-term economic decision, as evidenced by the 1918–1919 flu pandemic [32].

Given long pre-existing political attitudes towards "slow violence", enshrined through neoliberal policy prescriptions—not least of all the dismantling of public health capabilities in the name of responsible fiscal management and austerity—and the hegemonic evolution of widespread entrepreneurial subjectivity,[14] the decision to prioritize the economy over health should be seen as a consistent move by the police regime [33]. In fact, the police response has been brazen: the UK government's initial policy was that economically harmful lockdowns could be avoided if the nation would just accept the deaths of a few of the most vulnerable (which they confessed would still amount to a "very high death toll") while they worked towards herd immunity [29]; in the US, Texas Lieutenant Governor Dan Patrick even went as far as to claim that the elderly with grandchildren would willingly sacrifice their lives for the economic well-being of their progeny (and that they should) [30,34].

How, though, is a decision made regarding which lives are worth living and preserving? Amidst the totalitarian violence of the Nazi regime in mid-20th century Europe, the murder of mentally and physically handicapped people was justified according to a facile and ideologically tinged argument concerning the worthiness of life [8]. What has COVID-19 revealed about the ideological value of life today? It has demonstrated the hierarchy maintained by a police regime that is largely reducible to a (necro)police-economy. The elderly are to be inoculated or discarded precisely because they have the highest mortality rates and experience COVID-19 most severely; the economic bottom line is that this population is at risk of taking scarce hospital beds from young and productive workers who need to recover so as to keep the economic efficiency and competitiveness of society sustained [35]. For some, these policy decisions amount to elderly and even human rights abuses [36], but such resistance to the police regime is sparse compared to the well-organized and militant defense of committed policy decisions.

We should be asking why securitization—the mobilization of all state resources and the suspension of normal politics—has not taken place in the most affected nations. Necropolitics is not always about human-on-human violence (war, as such). In the COVID-19 pandemic, it is about exposure to death through activity identified (even if not acknowledged) as essential [8,18]. As commentators such as Michael Barnett have noted, the (neo)liberal international order is a sacrificial order based on inequality (despite the liberal pretense of equality, which for Rancière is a characteristic function of the police). Neoliberal capitalism has shaped our (un)ethical decisions as to whose life is of value and whose death is necessary for 'progress' [37]. The evidence is clear: ethnic minorities and other groups are disproportionally dying from COVID-19 and its related ailments [38,39]. This is a death sentence for the vulnerable and expendable, and yet the full force of the police regime—in this case the state—has not been mobilized. How many lives have been lost in the second, third, and perhaps even fourth waves of infection resulting from the Delta and other variants since the government made a conscious decision to prioritize the economy in the short term [40]? What we are witnessing is the necropolice economy engaging in an (un)ethics of exposure that is only exacerbating the crisis.[15]

## 4. The New Leviathan

*"It hardly makes any difference who will be the next president. The world is governed by market forces."*[16] [41]

In traditional political theory, sovereign authority holds a special place in relation to the boundary between the peaceful inside of the state and the violent outside of interstate relations. This is perhaps best illustrated by Carl Schmitt's pithy line: "Sovereign is he who decides on the exception" [42]. The boundary of law on Schmitt's account, as well as the definition of violence, is both determined and constantly re-evaluated by the sovereign power, whatever form it may take. Sovereignty is performed as such by the sovereign ability to decide where no codified guidance (e.g., the law) exists, or where existing guidance is inadequate to the task. In Schmitt's account, the exception is defined by a distinction between norms and decisions; thus, it is an important point out that what we have with COVID-19 is not a blatant flouting of the law, but rather a repeated act of (in)decision in relation to the subjugation of norms. It is the inability of neoliberally oriented states to be decisive in relation to the COVID-19 pandemic that reveals a shift in the power relations constituting sovereign authority.

One may be tempted at this point to invoke Giorgio Agamben as a counterpoint, in relation to both his general treatment of the state of exception as a political theoretical concept [43], and to his badly received comments concerning the outbreak of COVID-19 in Italy (wherein he denies any epidemic—let alone pandemic—reality underlying COVID-19, and accuses the Italian government of a power grab executed via the state of exception, which he believes has become a normal governing paradigm) [44]. However, Agamben's argument places too much weight on the role of the state as a centralized power structure

and too little on the role of the market as a decentralized (albeit institutionally protected) power structure driven by the necessities of capitalist accumulation.

As I have already argued, in weighing up the options, the sovereign state has found itself caught in a trap, a double bind in which to prioritize the economy exacerbates a public health crisis and to prioritize public health exacerbates an economic crisis. The UK has been facing a further economic quandary with its exit ('Brexit') from the European Union adding one more factor to the strategic calculus. In UK Prime Minister Boris Johnson's words: "We have to balance the risks of the disease and of continuing with legal restrictions, with their impact on people's lives and livelihoods." [45]. In the US, Fox News—which seemingly holds great sway over the former President, Donald Trump—ran segments entitled 'Flatten the curve, not the economy', imploring the government not to let the cure be worse than the disease: "You think it's just the coronavirus that will kill people? This total economic shutdown will kill people." [46]. It is not just the media that are putting pressure on the state to act in the interests of the economy, however. The emergence of a new group of intellectuals, exerting influence in particular through think tanks but also through the media, have been promoting a libertarian-authoritarian agenda that is ostensibly guided by a radical shift towards laissez-faire governance [29]. The assertion, to put it bluntly, is 'hands off the economy, and hands off public health as well' .

The sovereign state has, therefore, not so much been intent on a power grab, as Agamben would have it, but has rather been attempting to fulfil its role as guardian of the economic sphere. His claim is that terrorist 'alibi' justifying previous states of exception in the post-9/11 era, now exhausted, has been replaced by an epidemic (and pandemic) "pretext" [44]. Yet, this gives too much credence to the competence and aptitude of leaders working amidst an uncertain field of risk. Absolutely, as Agamben points out, leaders have exploited "states of collective panic", but there is no evidence that this has been part of a premeditated plan to unleash state power on the population; if anything, it has served as a pretext for states to bow down to the forces of capital. This places Agamben's argument about exception on a different register to my own. It is not the state that is exposing its exceptional authority, but the sovereign market forces to which the state must kowtow.

Amidst this pressure, the aesthetic that leaders have sought to present is one of competence—genius even—and of greater wisdom than the masses. However, in the context of strategy, leaders have been well aware that their actions are guided by probabilities of risk—that is, that their actions today are based on future potentialities. As Ulrich Beck has noted, risk can never be eliminated, only mitigated, and, in what he refers to as a "risk trap", every action taken to eliminate or mitigate a risk leads only to a new horizon of risk probabilities [47].

In weighing up the possibility of mass death of vulnerable populations against the possibility of economic crisis, there is no contest. Because capitalist accumulation through the market is inherently unstable, crises are a perennial risk. Marx explains this fragility through the delicate balance of circulation. Capital is, essentially, value in motion [48]. Capital, as money, is fed into the production process to produce commodities that are then sold for more money (initial investment plus profit) but must be reinvested into the process for that accumulated profit to remain as capital.[17] It is a snowball effect and goes a long way to explaining how and why growth is so crucial to capitalism. However, should money be hoarded—that is, not spent or invested—or the realization of profit be disrupted, then the whole cycle of circulation is disrupted, and the result is crisis. Lockdowns amount to almost certain crisis, particularly for those at the top of the class structure. President Trump, for all his flaws, understood this well. He knew that finance and risk are all about confidence—and by repeatedly downplaying the economic and health impacts, he was able to help sustain a booming finance market (a market that is able to suspend time and roll risk over) throughout the course of the pandemic, even as the actual economy figuratively burned (along with lives and livelihoods of hundreds of thousands of Americans).

Therefore, in medias res, leaders have been acting according to a different temporality—a temporality of contracting time. The time of COVID-19 is not linear—and certainly,

everyone hopes, not progressive. Action within the temporality of COVID-19 occurs with an approaching end in mind. According to this mode of thought, the pandemic will one day be over. It is a situation that might be likened to a reverse apocalypse: the coming eschaton in this instance being the exit from pestilence, rather than an approaching pestilence. We are, as such, in the eye of the apocalypse, looking for a way out.

For the prevailing regime it is, consequently, a matter of waiting. If only the necropolice economy can hold on, the pandemic will eventually end, the economy will boom, and everyone will go back to their allocated place. Risk can be solved according to this calculus, with only a little collateral damage incurred by disposable populations along the way. Society, as conceived alongside the imperatives of capital, can be renewed for the post-pandemic era. Death while-you-wait. As Mbembe notes, "[t]he present itself is but a moment of vision—a vision of the freedom not yet come". In the present crisis, these words take on a new meaning: freedom (from COVID-19) is yet to come, but there will be no justice to come.[18] Messianic thought is grounded in the notion of redemption, redemption for a blood price—but this blood price during COVID-19 is no price at all when the blood sacrificed is by those deemed expendable; those who must pay with their lives so that the rest of us may live—precisely whom Boris Johnson, Jair Bolsonaro, Dan Patrick, and others refer to. This is nothing but a selfish, secular messianism, wherein self-interested 'heroes' seek redemption for themselves at the cost of others. "Eat out to help out" has become a catchphrase, but who is being helped and who harmed [49]?

Amidst the period of contracting time, there has been mass opportunism by those insulated from vulnerability. Opportunism has been in full swing not only by the largely capitalist corporations, but also by capitalist agents of the state. Just as companies such as General Motors and Gilead have been able to exploit the situation and their position to secure exclusive lucrative contracts and immense profits, sweeping legislative changes imposed in places such as India upon citizens still in shock, suggests a move straight out of the disaster capitalism playbook [19,50]. As early as mid-2020, prior to the emergence of the Delta variant that has taken so many lives, Boris Johnson had the gall to say "I think it is great to see people out shopping again" [40], referring, of course, only to those who are still alive.

Plainly, what we are witnessing is state failure on a mass scale. The UK, US, and Brazil, in particular, have compounded the COVID-19 crisis through poor policy, imposing immense trauma upon their populations, as well as inducing a rising sentiment of generalized injustice [37]. In response to Bolsonaro's claim that "We just can't stand still, there is fear because if you don't die of the disease you starve", one street vendor simply replied "You're not going to die!" [51]. However, despite the prioritization of the economy, we have seen market failures as well. Market emergences can produce goods, for sure, but not always when required, as has been painfully obvious for not only PPE but basic goods as well throughout the pandemic. What has also become clear is that markets involve not merely an interaction between supply and demand, but also the signal of price—those who can and are willing to pay for things will always get what they need. Does anyone believe that the Warren Buffett's of this world will have had problems obtaining toilet paper?

All of this adds up to one thing: sovereign power is no longer solely in the hands of the state; it exists in a precarious modus vivendi with capitalist markets. Only by conjoining necropolitics with necroeconomics, as I have been attempting to do, can we hope to decipher the present crisis as it is still unfolding. The poor policy that has been on full show is a result of market compulsion. Cuts to public services such as the public health and welfare systems are economic decisions that reveal class divides: those that cannot afford healthcare are those who are not deemed productive—the incarcerated, the young and unemployed, the elderly who did not earn and hoard adequately throughout their lives; groups which all intersect with race and minority populations. These are all symptoms of structural processes related to neoliberal inequality and have resulted from decades of policy choices made in response to the power of capital [17]. This poor policy, in the form of long, ongoing dismantling of public health infrastructure, is just one way that

COVID-19 has played out [18]. Will public outrage reveal the emperor's lack of clothes? As I have argued above, necropolitics is a paroxysm occurring amidst a failing necropolice regime, and I would not be the first commentator to point out that the present crisis of neoliberalism has been a long time coming. Under the strain of a public health crisis it is not unimaginable to assume that social consent and capital accumulation are reaching a breaking point [29].

Yet, the fact that capitalism is a contradictory system has never hindered it in the past. Alarmingly, it has been its general catalyst towards resilient transformation. A Marxian understanding of capitalist development therefore reminds us that the necropolice-economy is not about intentionally nefarious actions but is consistent with the systematic operations of bloodthirsty capital [6,21]. Marx, as I have already noted, speaks of figurative death extensively in relation to the petty-bourgeoisie and capitalists in competition with one another, and although this is not the same as real death (putting aside worker exploitation leading to death), when put in the context of a crisis such as COVID-19 or climate change, we can see that the capacity to have a livelihood becomes a matter of life and death. The ruling classes who already have a livelihood of sufficient (and excessive) means are less likely to be exposed to the disease as are frontline workers or those living in cramped conditions.

The dialectical relation of capitalist progression then is consistent with the literal idea of death as a figurative invisible hand. In Warren Montag's account, "[d]eath establishes the conditions of life; death as by an invisible hand restores to the market what it must be to support life. The allowing of death of the particular is necessary to the production of life of the universal . . . it demands death be allowed by the sovereign power", [16] which is to say that parts of society must be exposed to death for the whole of society to persist and progress. This is precisely what is at stake in the necropolice economy: the market is dictating who is allowed to live and who must die, which can only mean the market has obtained sovereign status or, in what amounts to the same thing, control over the prevailing sovereign authority of the state. To once again quote Mbembe: "the ultimate expression of sovereignty resides, to a large degree, in the power and the capacity to dictate who may live and who must die. Hence, to kill or to allow to live constitute the limits of sovereignty, its fundamental attributes" [52]. Under a regime of necropolice economy, the preservation of capital and of productive over unproductive resources is absolutely facilitated by the state—such that in the contemporary world, the state is little more than the unfortunate lackey of a sovereign market.

The mystery of this sovereign market is the legitimacy of its authority. It offers no social contract, but rather demands that the sovereign state honor its own contract to not interfere in the private lives of its citizens. These citizens have given up their rights to the means and ends of violence in return for unimpeded freedom within the space of non-violence, a space that has historically become associated with society's economic function.[19]

Unlike the state, which derives its legitimacy from the threat or risk of violence and justifies its existence by keeping domestic violence at bay (maintaining the peace) and external violence away (through diplomatic and military means), the sovereign market has no justification beyond its own mimetic logic. That is, to act rationally in the market is to follow market price signals wherever they may lead. Knowing that a particular currency, for example, has no ultimate reason to drop in price,[20] but that there is a widespread fire sale in progress, necessitates the sale of that currency, not because it is justified by some rational end, but rather because it is justified on the basis of pure rational means, the logic of mimetic action.[21] Under this logic even irrational ends can be transubstantiated into rational action, for the market needs no reason except for its own movement in the pursuit of accumulation. In André Orléan's words: "In a market one does not act in accordance with what one believes, but with what the market believes." [53]. Amidst COVID-19, the necropolice economic function has followed this logic to its deadly extremes.

Because the market has self-referential justification, it confronts economic actors as an autonomous alien force that develops increasing disciplinary power over both individuals and the state[22]—hence the sovereign dilemma of exception amidst the pandemic [54]. The

distinction between state and market sovereignty is such that the state is articulated in terms of ends (the end of negating violence), while the market is articulated in terms of pure means (the means of accumulating private wealth and capital). Nevertheless, the justification of state sovereignty through the social contract is based upon an arbitrary delineation of what counts as violence. The state is conveniently permitted to designate violence as that which challenges state sovereignty—a circular logic that can only be explained by the distinction between politics and police (politics is violence, police is peace). Therefore, despite its obvious abnegation of the duty to banish violence amidst the COVID-19 pandemic, it is able to justify its authority through its role as guardian of the economic sphere, and with it the sovereign market.

The question relevant to an account of necropolice economy is why, given the historical ascendancy of the sovereign market (which reached near total domination with the fall of the Berlin Wall in 1989 and the subsequent collapse of the Soviet Union in 1990), the sovereign market has not moved to completely negate the sovereign state function—that is, why have even libertarian movements sought not to eliminate the state but rather to minimize the bureaucracy of the state function, thereby concentrating its power within a streamlined, less democratic, and more authoritarian regime?

The answer lies within the justificatory function of violence, because violence and death are and will always remain power's means of last resort. Market sovereignty is bestowed by no final end and is therefore self-legitimating. It cannot depend on violence directly, for that would contradict its own self-justification and reveal its own irrationality. It can, however, employ the state as an agent of action in both preventing and pursuing violence as economic protective measure.

That sovereignty confers the power to bestow both life and death is the central point of necropolitical analysis. The point of the necropolitical economy is that it puts the market in the role of decider—if millions must die to sustain production and consumption in the pursuit of profit, then that is as will be done [22]. Sovereignty also depends on violence; this is how it separates society from itself and offers 'protection' in exchange for rights and public goods; but it is a ruler that exploits its ruled—the market no less than the state. It is a protection racket just as violent as any gang from the Hollywood cultural imagination [7].

## 5. Conclusions

In the final days of 2020, experts at the World Health Organization warned that, as far as pandemics go, COVID-19 is "not necessarily the big one" [55]. Perversely, as a collective—species, society, class, whatever—we have been lucky. However, the notion of luck is a dangerous discursive fiction—it inures us to the traumatic memories that would otherwise sustain our horror and rage at the injustices delivered upon society. Just as we have become desensitized to the ubiquitous video footage of black people dying at the hands of militant and racist police officers (what Rancière refers to as the 'petty police'), so too are we at risk of becoming desensitized to the unnecessary death that surrounds us today [9,56]. For the 4.55 million people who are dead, along with their families and communities whose lives have been devastated by death and illness, luck has no discursive function. They bear the memory and trauma of this event as it continues to unfold.

In Kim Stanley Robinson's near-future fiction, *The Ministry of the Future*, he portrays a world in which climate change has arrived in full force—mass death from heatwaves, terrorist cells, quixotic and doomed technical band-aids. All the while, central banks, governments, and corporate executives deny the problem is theirs to prioritize or act upon. Nothing, it seems, is incentive enough for the machinations of capital to stop putting 'value in motion' [48]. Robinson's fiction is horrifying for its realism. What possible event might effect change? How deadly and devastating would a natural threat need to be before the world of capital is relegated to the needs of society? Fredric Jameson famously quipped that it is easier to imagine the end of the world than the end of capitalism.[23] However, for those who have died, it was and is the end of the world, and for those who have been

traumatized by the loss of loved ones, or by the indelible damages wrought upon their community, the end of the world is no quip.

We are in existential shock, and therefore ripe for shock therapy [30,50]. What comes next will depend on whether society accepts its dead as collateral damage, as a sort of evolutionary necessity, a social purification akin to the dystopian narrative of *Snowpiercer*; or whether society channels its grief and outrage into a revolutionary consciousness. Trauma as grief can be the catalyst for widespread challenges to elitist injustice, of the hypocrisy on show with no attempt to conceal it [37]. To focus on the sociopolitical effects of pandemic trauma might yet lead to a form of necropolitics and necropolitical economy that is for the people—an attempt to ursurp the necropolice regime that has prioritized capital over the lives of the vulnerable without replacing the injustice with a new police regime and more capitalist injustice. What some commentators have referred to as necroresistance [57], and others as thanatopolitics [58], does not quite capture what is at stake here, for those theories tend to emphasize death in terms of volition—death as an intentionally political act, rather than death as a politicizable event. We are in the midst of a world-historical moment, a moment of revelation in the literal sense of revealing that which is hidden, of making visible that which had previously been invisible, of disrupting the police function and inaugurating a new horizon of justice [10]. Confrontation with mortality might aggravate death and vulnerability disavowal strategies, but it also might lead to collective solidarity and emancipatory action [30]. To make visible the inequality within the designated field of police is to burst its legitimacy asunder; to promote a situation—pace Rancière—wherein everyone counts, not just those who are instrumental to the vagaries of capital [9]. The expendable dead during the COVID-19 pandemic must be memorialized, but not in the way unnamed soldiers are—as a nationalist symbol of unity that serves only the police regime; rather, in a way that allows the narrative to be controlled by those who demand it never be repeated [59]—by those who would challenge the necropolice order and transform it into a necropolitical response that can end the reign of necropolitical divisions. Public health emergencies may well be unavoidable, but the economy can be prioritized in a way that does not privilege market power and distortions; for all of the biopolitical excesses of a country such as China, it acted swiftly to protect its citizens not only by containing the threat of the virus, but also by offering sustenance so as to sustain the material reproduction of life [22]. We must learn the lessons of countries such as South Korea, who after SARS prepared and invested heavily to ensure future events will not repeat the mistakes of the past. If COVID-19 really isn't 'the big one', we can afford to do nothing less.

**Funding:** This research received no external funding.

**Conflicts of Interest:** The author declares no conflict of interest.

## Notes

1. Note that it is not a sufficient cause that their productivity be essential, it is also necessary that there be a surplus of individuals (reserve army of labor) who may replace them in their laboring activities [6].

2. As will be outlined in what follows, I take my cue on this point from Jacques Rancière [9].

3. Mark Howard has focused on the function of the event in inaugurating new, imperfectly formed, conceptions of justice that are always doomed to failure [10].

4. Refs. [13,14] Clausewitz's original phrase 'war is continuation of politics by other means' was intended to highlight that war must remain an instrument of policy, and that it is imperative to keep the political objective in focus whenever engaging a war, for otherwise the war may come to drive itself as a means without an end. On Foucault's account, this formulation only becomes possible with the advent of the sovereign state, because it is only then that war can be centrally commanded and pushed to the boundary of sovereign space as a means of keeping others at bay.

5. Foucault uses the examples of the Romans in France and the Normans in England to make this point, but I also could point to the enduring need to reassert the union of the United States and the recurring themes of division between states historically designated as Confederate and Unionist.

6. Rancière refers to this as 'consensus democracy' so as to distinguish it from a true 'democracy' pertaining to the insurgent act [9].

7    What is at stake for Marx is the capitalist's survival as a capitalist; to fail as a capitalist would be to enter a new class role in society. cf. [21].

8    The struggle for legal limits on the working day is nothing less than a struggle through which workers can be saved "from selling themselves and their families into slavery and death" (p.416) cf. chapter 10, 'The Working Day' [6].

9    It is of course debatable whether capitalists are themselves productive; this is one critique given by the socialist position.

10   The unconditionality of death amongst non-productive members of society would be permissible for only as long as the sovereign body maintains its legitimacy; should mass death challenge the ruling (police) regime, policy change would be expected.

11   For instance, through misguided attempts to achieve 'herd immunity'.

12   Specifically, Foucault says it becomes a principle that the government must not act directly on the economic process, but may only intervene in favor of it, cf. [12].

13   As per Rancière, it is only where equality is denied that policing takes place, and it is this situation out of which politics must emerge. Both the state and market sovereign rely on inequality and deny equality [9].

14   Treating the self as a business, as an entity that must be invested in, and marketed, and 'sold' is considered a characteristic feature of neoliberalism by Foucault (and Brown, following Foucault). cf. [12,33].

15   Cf. in particular comments by Margot Kushel and Sharelle Barber in [5].

16   Alan Greenspan, then Chairman of the Federal Reserve (Fall 2007), quoted in [41].

17   This is Marx's famous 'General Formula for Capital', M-C-M', cf. [6].

18   I derive this notion from the claim that every 'world-historical moment' is characterized by a vision of justice presented as the figurative 'end of history', a figuration that is suggested to be little more than a recurring fiction. The COVID-19 pandemic is peculiar in that its vision of justice seems oriented towards not the end of history, but the end of this moment in history so that a new 'world-historical moment' may begin. It is, as such, a self-aware moment in history. Cf. [10].

19   Specifically, Foucault says it becomes a principle that the government must not act directly on the economic process but may only intervene in favor of it. Cf. [12].

20   Here we must leave aside the question of value, for price operates according to a logic of its own. For André Orléan, to whom my analysis here is indebted, value does not precede the act of exchange. To suggest that it does is to commit to the unsustainable claim that individual economic actors are themselves sovereign in knowing exactly what (utility) they want in all circumstances.

21   Here we must leave aside the question of value, for price operates according to a mimetic logic of its own. For André Orléan, to whom my analysis here is indebted, value does not precede the act of exchange. To suggest that it does is to commit to the unsustainable claim that individual economic actors are themselves sovereign in knowing exactly what (utility) they want in all circumstances. Orleán's aim is to dismiss any theory of value that does not rely on the mimetic effect, including theories of utility, labor (i.e., Marx) and scarcity (e.g., Hayek) Cf. [53].

22   Marx develops this point in more detail in the famous 'Fragment on Machines' cf. [54].

23   Mark Fisher. *Capitalist Realism: Is There no Alternative?* (John Hunt Publishing, 2009), p. 7.

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
