# Peer review of "The Necropolice Economy: Mapping Biopolitical Priorities and Human Expendability in the Time of COVID-19"

_societies, doi:10.3390/soc12010002_

Round 1

Reviewer 1 Report

This is a very thought-provoking account of the necropolitics of Covid thorough the prism of wide-ranging and well-selected theories including Marx, Ranciere, Heidegger to Foucault. I share the sentiment of many of the points raised and I am pleased to see that the author(s) identify neoliberal capitalism as the main problem and not, for example, populism. I recommend this essay for publication as it is.  

Author Response

Dear Reviewer,

Thank you for your generous approval of my paper, I am glad you liked it. 

For completeness, let me enumerate some edits I have made in response to another reviewer's feedback:

1) Justification of Rancière as a biopolitical thinker: You will see a new paragraph inserted between pp.3-4 outlining my justification for use of Rancière's distinction of politics/police in terms of necropolitics/biopolitics. I have also made minor updates throughout the paper highlighting the notion of visibility, which is associated with both the police and the field of biopolitics.

2) Acknowledgement of Agamben’s intervention: While I appreciate Agamben’s work on the state of exception as a political concept, like almost everyone else I do not subscribe to his musings on the state of exception as it pertains Italy’s (or any other country’s) COVID-19 response. Consequently, I do not believe the state of exception is particularly relevant to my argument concerning the necropolice-economy. If anything, I would argue that a surprising feature of the pandemic is that this has not happened (cf. my comments on the absence of securitization p.8). Nonetheless, it makes sense to engage Agamben on this point and I have done so on pp.8-9. I have also included some comments on p.8 as to why I think Schmitt’s account of exception makes more sense in relation to the argument concerning the necropolice-economy.

3) Explaining the focus on Capitalist Markets: Despite the fact there are few (if any) non-capitalist economies left standing in the world today, it makes sense to offer a brief explanation of why capitalism (and by implication a few choice neoliberal states at the vanguard of capitalist reproduction) i central to my argument concerning the necropolice-economy. You will find this justification on pp.4-5, where I argue that it is the substitution of society as a whole for society as a productive mass that makes capitalism a distinctive system in terms of the necropolice.

4) Engagement with Koselleck and question of ‘avoiding’ pandemics: I was not entirely clear on this point. The reviewer suggests that my invocation of Koselleck was for the purpose of highlighting his thoughts concerning eschatology, but – although that might be a good point! – my intention was merely to use his work on memorialization to make my own point about the distinction between regime-led memorialization (which under the necropolitical-economy serves capital and the productive members of society only) and a society-led memorialization (which would serve society as a whole). Also, the reviewer makes a comment (that I could not fully connect to the comments on Koselleck) about pandemics being combatable, but not avoidable. I agree, and have made a minor edit on p.14 expressing the point that while public health emergencies may be unavoidable, economic prioritization at the expense of society as a whole is not. N.B. I opted here for the term ‘public health emergencies’ rather than ‘pandemics’ as I think it is arguable (though not in the space permitted) that pandemics are exacerbated by the interconnected movements of people engaged in economic activities under globalized capital – if the growth cycle was not as dependent on globalized sourcing efficiencies then pandemics may, at a minimum, be limited to epidemic status.

5) Focus on relationship between thanatopolitics and biopolitics: Here, I found myself somewhat confused as to whether the term ‘thanatopolitics’ was introduced as a distinctive theoretical approach (as outlined in the work of SJ Murray, for instance), or as another term for necropolitics. Accordingly, I have tried to address the comment as both a request to focus on the relationship between necropolitics and biopolitics (particularly in relation to my justifications regarding Rancière, as per point 1), AND by introducing the concept of thanatopolitics as it appears in the literature and of distinguishing my argument from it. For the latter point, cf. p.13, wherein I make the distinction between necropolitics as a positive politicization of death as an already-occurred fact, and death as a present and/or future act intentionally undertaken. For the former point, see my comments on pp.3-4, 7, 13.

Thank you very much for your time and consideration in reviewing my paper.

Reviewer 2 Report

The author submitted an article that addresses an expanding domain. Core-bibliography develops as we live and write, and therefore, any research submitted on COVID topics will be a core-resource for any other scholars from this field. I appreciate the economical framework that tackles adapting biopolitics to a so called thanatoeconomics. I do have some recommendations, though, that could inforce the sustainability of this research:1. The author should introduce a little paragraph in order to explain on what grounds Ranciere, who is not an author of biopolitics, has been engaged here and to what extent his distinctions on police and politics are susceptible of a biopolitical potential. 2. There is a well known debate between Agamben and Nancy, on the effects of COVID-19 pandemics as a state of exception on current democracies. The state of exception should be better addressed by direct references to Agamben, who is absent from the list of references, and major polemics such as their exchange of arguments should be used in order to tackle the so-called biopolitical priorities. 3. It is not clear why the author restricts the realm of this research solely on capitalist markets. He/She said that: `Third, I argue that the inability of states to be decisive in the pandemic reveals that the sovereign prerogative to decide on the exception is constrained by capitalist forces`. Pay attention to the fact that we deal with a universal state of exception - this is the nouance of a pandemic - and therefore, it impacts areas where capitalism is not a fundamental ideology - see equally the situation of communist or underdeveloped countries across the world. 4. In the end, the author engages Koselleck, most probably for his idea that modernity faces constantly, regardless its historical phase, an appetite for a eschatological approach. ("The expendable dead during the COVID-19 pandemic must be memorialized, but not in the way unnamed soldiers are – as a nationalist symbol of unity that serves only the police regime; rather, in a way that allows the narrative to be controlled by those who demand it never be repeated– by those who would challenge the necropolice order and transform it into a necropolitical response that can end the reign of necropolitical divisions"). To prevent from happening - that could be something as a moral awareness that Adorno engaged, for example, in addressing the cause of the Holocaust in order to avoid the repetition of such traumatic history. But in case of pandemics, avoiding seems rather not in our power.  To combat yes, but to avoid, no. 5. Last but not least, I recommend a more focused approach on the relationship between thanatopolitics and biopolitics (inspired by Foucault) conceived at a biopolitical level and on the immediate correspondence of those two constructs at an economical level. If thanatopolitics would have been engaged into your argument, would the opinions on the dynamics of economy in times of pandemic radically change?

Author Response

Dear Reviewer,

Thank you for your generous feedback on my paper. 

Let me enumerate some of the edits I have made in response to your feedback:

1) Justification of Rancière as a biopolitical thinker: You will see a new paragraph inserted between pp.3-4 outlining my justification for use of Rancière's distinction of politics/police in terms of necropolitics/biopolitics. I have also made minor updates throughout the paper highlighting the notion of visibility, which is associated with both the police and the field of biopolitics.

2) Acknowledgement of Agamben’s intervention: While I appreciate Agamben’s work on the state of exception as a political concept, like almost everyone else I do not subscribe to his musings on the state of exception as it pertains Italy’s (or any other country’s) COVID-19 response. Consequently, I do not believe the state of exception is particularly relevant to my argument concerning the necropolice-economy. If anything, I would argue that a surprising feature of the pandemic is that this has not happened (cf. my comments on the absence of securitization p.8). Nonetheless, it makes sense to engage Agamben on this point and I have done so on pp.8-9. I have also included some comments on p.8 as to why I think Schmitt’s account of exception makes more sense in relation to the argument concerning the necropolice-economy.

3) Explaining the focus on Capitalist Markets: Despite the fact there are few (if any) non-capitalist economies left standing in the world today, it makes sense to offer a brief explanation of why capitalism (and by implication a few choice neoliberal states at the vanguard of capitalist reproduction) i central to my argument concerning the necropolice-economy. You will find this justification on pp.4-5, where I argue that it is the substitution of society as a whole for society as a productive mass that makes capitalism a distinctive system in terms of the necropolice.

4) Engagement with Koselleck and question of ‘avoiding’ pandemics: I was not entirely clear on this point. The reviewer suggests that my invocation of Koselleck was for the purpose of highlighting his thoughts concerning eschatology, but – although that might be a good point! – my intention was merely to use his work on memorialization to make my own point about the distinction between regime-led memorialization (which under the necropolitical-economy serves capital and the productive members of society only) and a society-led memorialization (which would serve society as a whole). Also, the reviewer makes a comment (that I could not fully connect to the comments on Koselleck) about pandemics being combatable, but not avoidable. I agree, and have made a minor edit on p.14 expressing the point that while public health emergencies may be unavoidable, economic prioritization at the expense of society as a whole is not. N.B. I opted here for the term ‘public health emergencies’ rather than ‘pandemics’ as I think it is arguable (though not in the space permitted) that pandemics are exacerbated by the interconnected movements of people engaged in economic activities under globalized capital – if the growth cycle was not as dependent on globalized sourcing efficiencies then pandemics may, at a minimum, be limited to epidemic status.

5) Focus on relationship between thanatopolitics and biopolitics: Here, I found myself somewhat confused as to whether the term ‘thanatopolitics’ was introduced as a distinctive theoretical approach (as outlined in the work of SJ Murray, for instance), or as another term for necropolitics. Accordingly, I have tried to address the comment as both a request to focus on the relationship between necropolitics and biopolitics (particularly in relation to my justifications regarding Rancière, as per point 1), AND by introducing the concept of thanatopolitics as it appears in the literature and of distinguishing my argument from it. For the latter point, cf. p.13, wherein I make the distinction between necropolitics as a positive politicization of death as an already-occurred fact, and death as a present and/or future act intentionally undertaken. For the former point, see my comments on pp.3-4, 7, 13.

Thank you very much for your time and consideration in reviewing my paper.

I look forward to your next response.